# Polyvictimization and Adolescent Health and Well-Being in Ethiopia: The Mediating Role of Resilience

**DOI:** 10.3390/ijerph20186755

**Published:** 2023-09-13

**Authors:** Lior Miller, Nicole M. Butera, Mary Ellsberg, Sarah Baird

**Affiliations:** 1Department of Global Health, Milken Institute School of Public Health, The George Washington University, Washington, DC 20052, USA; mellsberg@email.gwu.edu (M.E.); sbaird@gwu.edu (S.B.); 2The Biostatistics Center, Department of Biostatistics and Bioinformatics, Milken Institute School of Public Health, The George Washington University, Washington, DC 20052, USA; nbutera@bsc.gwu.edu; 3Global Women’s Institute, The George Washington University, Washington, DC 20052, USA

**Keywords:** adolescence, physical health, Ethiopia, gender, mental health, path analysis, polyvictimization, resilience

## Abstract

Interpersonal violence is a pervasive experience affecting one billion children and adolescents annually, resulting in adverse health and well-being outcomes. Evidence suggests that polyvictimization, the experience of multiple forms of violence, is associated with more harmful consequences for adolescents than experiencing individual types of violence, although data from low-and middle-income countries are limited. This study analyzed data on over 4100 adolescents from the Gender and Adolescence, Global Evidence Study in Ethiopia to examine the association between polyvictimization and adolescent mental and physical health and the mediating role of resilience using linear regression and path analysis. We hypothesized that adolescents experiencing polyvictimization would experience worse mental and physical health than those experiencing no types or individual types of victimization, and that resilience would mediate these relationships. Half of sampled girls and over half of boys experienced polyvictimization. Among both sexes, polyvictimization was associated with worse mental but not worse physical health. Resilience mediated the association between polyvictimization and mental health among girls only. Strengthening resilience among girls may be an effective avenue for mitigating polyvictimization’s negative mental health effects, but additional research and programming for preventing and identifying polyvictimized adolescents and linking them to care is needed.

## 1. Introduction

Violence against children and adolescents is a serious and pervasive public health, development, and human rights issue affecting an estimated one billion children aged 2–17 each year [1,2]. In Ethiopia, an estimated 75% of children and adolescents ages 4 to 18 years old report experiencing physical or psychological violence at home and at school, and over 50% of children and adolescents experience community violence [3]. These figures are likely an under-estimation, given that they only include more commonly studied types of victimization such as physical violence, emotional violence, sexual violence, and bullying, but do not include witnessing domestic violence, child marriage, female genital mutilation/cutting (FGM/C), or dating or intimate partner violence, which are sometimes recognized as forms of victimization and/or adverse childhood experiences (ACEs) with deleterious consequences [4,5,6]. Further, studies usually focus on single forms of violence and one single type of location or perpetrator [1]. Few studies assess the multiple forms of violence, perpetrators, and settings where violence occurs [1]. Indeed, multiple definitions of victimization, sometimes referred to in the literature as child maltreatment, with some including only forms of physical, emotional, or sexual victimization [7]. More comprehensive definitions of victimization also include peer and sibling victimization, experience of conventional crime, indirect victimization such as witnessing family or community violence [8,9].

The construct of polyvictimization, or the experience of multiple types of violence, takes into account a more comprehensive understanding of violence, recognizing that children and adolescents can be victimized in multiple ways and across multiple settings and by multiple perpetrators, and that these experiences of victimization are often connected [8,10,11]. Children and adolescents exposed to one form of violence are more likely to experience other forms, reflecting how victimization experiences co-occur and interact, and leading some researchers to characterize polyvictimization as a condition rather than as a series of events [12,13]. Further, there is evidence that polyvictimized children and adolescents, compared to other victimized children and adolescents, experience high levels of more serious forms of victimization such as sexual abuse or sexual assault [14,15].

This study uses quantitative data from the Gender and Adolescence, Global Evidence (GAGE) study in Ethiopia, to assess the prevalence of polyvictimization, its association with mental and physical health outcomes using linear regression, and the potential mediating role of resilience using path analysis. The study makes a novel contribution to previous studies on polyvictimization, by including forms of victimization, including gender-based violence and harmful traditional practices such as FGM/C, not traditionally included in its measurement. It is the first study assessing polyvictimization in Ethiopia of which we are aware.

Research on polyvictimization is informed by cumulative risk theory, which posits that the more risk factors an individual is exposed to, the greater the likelihood of a negative outcome in a dose-response effect [16,17,18]. The forms of violence commonly included in definitions of polyvictimization are physical abuse, emotional abuse, sexual abuse, neglect, exploitation, victimization by peers and siblings, witnessing family and community violence [19,20]. Some scholars categorize witnessing domestic violence as an ACE instead of a form of child maltreatment or violence against children, due to the potential risk of blaming mothers for exposing their children and mothers losing child custody under failure to protect legislation and policies, even when fathers are more likely to be the perpetrator [21,22]. In addition, there is often conceptual vagueness in how witnessing domestic violence is defined, and the thresholds for intervention [21]. We include witnessing violence as an indirect form of victimization following other research on polyvictimization [8] and the United Nations’ definition included in the 2006 World Report on Violence Against Children [23]. Conventional crimes such as robbery and physical assault, and cyber bullying [8,10]. While polyvictimization is a universal issue, affecting adolescents in both high-income countries (HIC) and low-and middle-income countries (LMICs), it may manifest differently in different country and cultural contexts [10,24]. Some studies in LMICs have explored how FGM/C [24] and child marriage [25] both forms of harmful traditional practices, are associated with experiences of polyvictimization and violence, but have not included these forms of gender-based violence in their measurement of polyvictimization. Other studies in LMICs also include forms of school-based violence such as corporal punishment [26,27], which is a rarer form of violence in high-income countries (HICs) where the concept of polyvictimization was originally developed. However, school violence is common in many LMICs, and it is a form of violence that often affects boys more than girls [28]. We include FGM/C as a common harmful traditional practice that is also considered to be a form of gender-based violence in Ethiopia and other LMICs directed towards girls [29], and school violence, as common types of victimizations in LMICs that are important for understanding the gendered experiences of different forms of victimization [30].

Risk factors for experiencing polyvictimization span each sphere of the social ecological model [13]. Individual level risk factors for polyvictimization include school enrollment (although the evidence is mixed; in some settings it is protective and in others, it is a risk factor) [31,32] older adolescent age [31], having a disability [33], being an orphan [34], the ways and places an adolescent spends their free time (in particular, spending more time alone, in public spaces, or with a dating partner), and adolescent alcohol use, drug use, or smoking [13]. The types of violence that adolescents experience may also be gendered. In Ethiopia, the available data indicate that boys are more at risk of experiencing physical violence from caregivers, violence from teachers, and violence from peers [30], and that girls are at greater risk of experiencing forms of gender-based violence, including FGM/C, child marriage, and sexual violence [30], although research and analyses on violence against boys in LMICs is more limited than research on girls [35]. Adolescence itself, a period marked by significant developmental, sexual, behavioral, and social changes [36], may increase vulnerability to specific forms of violence perpetrated by their parents or caregivers, peers, intimate partners, or teachers [37]. However, violence against adolescents, especially how such violence is gendered and affects girls and boys differently, remains an under-explored area in research, policies, and programming.

At the interpersonal level, polyvictimization risk factors include family structure (e.g., living with a non-biological father or in a single-headed household), household poverty, parental history of substance use, parental illness or psychiatric problems, parental intimate partner violence (IPV), and quality of relationships (both family and peer) [13,31,38,39,40]. Community, societal, and structural polyvictimization risk factors include neighborhood-level poverty, living in unsafe neighborhoods, mistrust and low social cohesion, gender and social inequalities, weak legal sanctions, and gender and social norms that support violence or foster conditions in which violence may be more or less likely to occur or to be condoned [38,41,42,43]. Inequitable gender norms in particular are significantly associated with experiencing physical and or psychological household violence [37].

The importance of polyvictimization for understanding the effects of violence stems from a growing body of research that demonstrates that the greater the number and types of victimization experienced, the higher the risks of adverse outcomes [39], in particular, for health and well-being using the WHO’s definition as a “complete state of complete physical, mental, and social well-being and not merely the absence of disease” [44]. For example, children and adolescents experiencing polyvictimization face greater risks of poor mental and physical health and emotional, behavioral, and academic consequences than those who experience discrete forms of violence or none at all, due to the cumulative effects of polyvictimization [10,45,46]. However, the literature on the mental health and behavioral health impacts of polyvictimization on adolescents is more extensive compared with the literature on physical health or sexual and reproductive health outcomes [47]. While there are fewer studies examining the effects of polyvictimization on adolescents in LMICs compared with HICs, the evidence base is growing, and demonstrates that polyvictimization is associated with adverse mental health outcomes including suicide ideation, post-traumatic stress, depression, anxiety, and low self-esteem [10,48]; increased health risk behaviors such as initiating violence and alcohol use [10,49]; reduced health-related quality of life [50]; and developmental problems including compromised cognitive, intellectual, behavioral, and emotional functioning [46]. Further, particularly egregious forms of victimization, such as sexual victimization and child maltreatment, have a greater impact on trauma symptoms [15], suggesting that some types of polyvictimization may be more severe than others. Evidence suggests that there may also be gendered differences on the impacts of polyvictimization on girls and boys. Adverse mental health outcomes may be more prevalent among girls that experience polyvictimization compared to boys, with polyvictimized girls experiencing increased likelihood of depression, perceived stress, suicidal thoughts, and post-traumatic stress [51,52]. In contrast, polyvictimized boys are more likely to self-report negative health or to experience more pronounced negative health effects than girls [52,53].

Given the evidence linking polyvictimization to adverse outcomes, a recent scoping review identified the need for more polyvictimization research on positive coping strategies and on resilience in particular, in order to understand its role in strengthening mental resources among polyvictimized adolescents and mitigating adverse health and mental health outcomes [54]. Resilience refers to positive adaptation and coping in spite of significant adversity or hardship [55,56]. Studies on polyvictimization and ACEs suggest that resilience mediates the association between victimization and adverse mental health outcomes such as post-traumatic stress disorder (PTSD) and depression in adolescence and physical health outcomes in adulthood [56,57,58,59,60]. For example, one study found that resilience reduced the strength of the relationship between victimization and traumatic symptoms among victimized adolescents sampled in a clinical setting [56]. Therefore, resilience may have a buffering effect on the negative sequelae of polyvictimization. However, few studies have investigated if resilience mediates the association between polyvictimization and negative health outcomes in adolescence, and no studies were identified that investigate this association in an LMIC such as Ethiopia, highlighting a research gap.

We examine the impact of polyvictimization on adolescent mental health and physical health, whether resilience mediates this relationship, and whether there are any differences in these relationships by sex. We hypothesized that polyvictimized adolescents would self-report worse mental health and physical health outcomes than adolescents self-reporting single forms of violence or none at all. We also hypothesized that resilience would mediate the relationships between polyvictimization and mental health and physical health.

This study contributes to the nascent research exploring how polyvictimization affects self-reported health outcomes among adolescents in LMICs and adds to the existing literature on mental health outcomes. It is the first study that we are aware of carried out in Ethiopia on polyvictimization and that examines the mediating role of resilience in an LMIC setting. By considering how polyvictimization affects adolescent boys and girls differently, examining how individual, community, and societal-level factors shape polyvictimization and adolescent well-being outcomes, and exploring a potential mediator of the victimization-adolescent well-being relationship, this paper contributes to more integrated, holistic approaches to inform violence prevention and to support healthy adolescent development.

## 2. Materials and Methods

### 2.1. Study Design

The study data came from the Gender and Adolescence: Global Evidence (GAGE) study. GAGE (2015–2024) is the largest global study on adolescents, following 20,000 girls and boys in six LMICs (Bangladesh, Ethiopia, Jordan, Lebanon, Nepal and Rwanda) to understand what works to enhance adolescent capabilities and empowerment [61]. The study’s conceptual framework [62] builds on the social ecological model [63] and the Child-Centered Integrated Framework for Violence Prevention [64], in order to account for how polyvictimization is driven by multiple levels of the social ecology, including individual risk factors, gender norms, and wider social, political and economic systems that shape and contribute to the conditions in which violence occurs [65]. The Child-Centered Integrated Framework for Violence Prevention identifies three categories of risk factors (individual, interpersonal, and community-level factors) and two categories of drivers (institutional and structural level) which place the individual in a series of overlapping circles [64]. This framework emphasizes how risk factors and broader societal drivers interact with each other; how these risk factors change based on the individuals’ identity, developmental stage, and social context; and how these risk factors and drivers are complex and dynamic. Unlike the original social ecological model, the framework situates the individual at the center of this graphical representation rather than being in the smallest of a series of concentric circles [43,64]. For this study, the framework was adapted by centering an adolescent, rather than a child, in the model (Figure 1), and it was used to guide the development and interpretation of this paper’s analytical models, including the selection of independent variables and covariates.

For this study, we used two rounds of data from GAGE in Ethiopia, the baseline and round two surveys. The baseline survey was carried out in late 2017 through early 2018, and the second survey was from November 2019 to February 2020 [67]. The quantitative surveys were administered to adolescents and the adult female caregiver/guardian. The study uses data from six locations in three regions of Ethiopia including Afar, Amhara, and Oromia and from Dire Dawa city administration (Figure 2) [68]; these sites comprise a mix of rural and urban communities. The data include adolescents from 175 rural and 19 urban kebeles (sub-districts) that were randomly sampled. For more information on sampling see the GAGE Ethiopia quantitative research design and sample manual [68].

Ethiopia is the second most populous nation in Sub-Saharan Africa with more than fifty percent of its population under 20 years of age [70]. A low-income country, it has seen major reductions in its poverty rate and improvements on other human development indicators, although significant challenges remain [71]. Despite this progress, violence against children and adolescents is pervasive in Ethiopian homes, schools, and community settings [3,30]. In addition, political instability and conflict in Ethiopia, including in GAGE study sites, have led to forced displacement and increased exposure to violence among children and adolescents [72]. The government of Ethiopia has made policy efforts to address violence against women, children, and adolescents, including the development of a five year national strategy on violence against women and children [73], a national policy on gender-based violence (GBV) prevention and response, a national children’s policy [74], and a national strategy and action plan on harmful traditional practices against women and children in Ethiopia [75]. It has also criminalized FGM/C [70] and child marriage [75] and developed a national costed roadmap to end child marriage and FGM/C [76]. Population-based data on violence against children and adolescents are scarce, however, and no studies on polyvictimization and its impacts on adolescent health and mental health have been carried out in Ethiopia to date; previous studies have primarily focused on the psychosocial consequences of child sexual abuse in clinical samples [77,78].

### 2.2. Sample

The sample used in this analysis comprises never-married adolescent girls and boys aged 10–12 years at the time of the baseline survey, who were randomly sampled using a census-style listing exercise [68]. In addition to random sampling, purposive sampling at baseline was carried out in order to target particularly disadvantaged adolescents, including those with disabilities, who are included in this study’s sample. Married adolescents (n = 69) were excluded from this analysis because never-married adolescents are likely to experience a different set of violence risk factors than married adolescents in the Ethiopian context. For example, never-married adolescents are more likely to live at home and to be enrolled in school, increasing their exposure to household violence and school-based violence, whereas married adolescents would be more likely to be exposed to IPV. The final sample includes 4106 adolescents, including 2263 females and 1843 males.

### 2.3. Measures

This study used data from a subset of modules in the adolescent and adult female surveys. All surveys were translated into local languages (Amharic or Afaan Oromo) and field tested [79]. Data were collected by trained enumerators via face-to-face interviews. Enumerators were generally from the same region and the same sex as the adolescent [79].

The main predictor variable is polyvictimization. A binary measure was created, using round two survey data, representing whether or not an adolescent reported experiencing two or more different types of victimization in the last 12 months prior to the round two survey data collection. All questions on victimization asked about the experience of victimization in the past 12 months with the exception of sexual victimization, which included options for reporting both victimization in the past 12 months and in the adolescent’s lifetime. The experience of lifetime victimization was collected for sexual victimization questions at round two; however, these lifetime victimization data were not used in the polyvictimization measure for consistency purposes in this paper’s analyses. Adolescents who reported experiencing two or more types of victimization (out of seven possible types) at round two of the survey were considered to be polyvictims, while adolescents who experienced either zero or one type of victimization only were considered to be non-polyvictims. The seven individual types of violence included in the polyvictimization measure included household physical violence, household emotional violence, sexual violence, witnessing violence against the mother/female guardian, (this included either violence perpetrated against the adolescent’s mother/female guardian by the adolescent’s father/male guardian, or by another family member), peer violence, school violence, and FGM/C (Table 1). Appendix A provides the full text of the survey items including the acts included in each item [80]. Due to the sensitive, socially undesirable nature of reporting witnessing violence, adolescents who reported the refused option for these questions were assumed to have witnessed violence, but the number of refused responses in the sample was minimal (five responses, or 0.12% of the sample). We also ran these regressions dropping any refused responses from adolescents, and the results were consistent. Six items on peer violence were adapted from the international version of the Trends in International Mathematics and Science Study (TIMSS) 2015 Grade 4 Student Questionnaire [81] to assess peer violence. For FGM/C, adult female survey responses were compared to the adolescent responses, and in cases where the adolescent responded do not know and the adult female decision-maker responded yes, the adolescent was scored as having experienced FGM/C.

All covariates used in the regression analyses and path analyses are described in Table 2.

Disability was assessed through a series of ten questions on physical functioning from the Census Questions on Disability endorsed by the Washington Group [83]. Household poverty was assessed using a combination of 16 items from the Ethiopia Demographic and Health Survey, the Living Standards Measurement Study survey, and the global Multidimensional Poverty Index [70,84,85]. The asset deciles were standardized within rural versus urban locations, in order to more closely approximate relative poverty, which has been demonstrated to moderate adverse health outcomes among maltreated children, even when controlling for absolute poverty [86]. Cronbach’s alpha was used to measure the reliability of the attitudes and norms measures, with values in the 0.70–0.95 range considered acceptable for this measure and all subsequent measures, based on the literature [87]. Cronbach’s alpha for the gender attitudes items was α = 0.954. Cronbach’s alpha for community gender norms was α = 0.692, close to the range of acceptability. The covariates were selected based on the relevant literature, the study’s conceptual framework, and theory in order to separate the relationship between polyvictimization and mental health and physical health from other factors and to reduce potential confounding [15,26,37,53,79].

The Child and Youth Resilience Measure-12 was used to assess resilience among adolescents. This 12-item scale is a measure of youth resilience that has been validated across multiple cultures and countries [88,89]. The items assess four components of social ecological resilience including individual, relational, community, and cultural aspects [88]. Item response options are yes, sometimes, or no. Previous factor analyses suggest a single factor solution [90]. The answers to the items were summed to create a resilience score, where greater resilience is indicated by higher scores. The Cronbach’s alpha for the 12 items was α = 0.768.

Adolescent mental health was assessed by the General Health Questionnaire-12 (GHQ-12) [91]. The GHQ-12 is a 12-item scale used to screen non-psychotic and minor psychiatric disorders in the general population and has comparable psychometric properties to longer versions of the GHQ [92]. It is suitable for use with adolescents and adults. It focuses on the respondent’s current mental state [93]. Binary scoring was used with 1-0-0-0 for positive statements and 0-0-1-1 for negative statements. Scores were summed ranging from 0 to 12, with higher scores indicating more adverse mental health outcomes. The Cronbach’s alpha for the 12 items was α = 0.875.

Overall health status was assessed using the following question adapted from the Health Information National Trends Survey (HINTS-5) [94]: “In general, would you say your health is very good, good, fair, poor or very poor?” A dichotomous indicator was created equal to 1 for fair, poor, or very poor health and equal to 0 otherwise.

The GAGE study research design and tools received approval from the George Washington University Committee on Human Research’s Institutional Review Board (071721), the Overseas Development Institute Research Ethics Committee (02438), the Ethiopian Development Research Institute (EDRI/DP/00689/10), the Addis Ababa University College of Health Sciences Institutional Review Board (113/17/Ext), and the Afar, Amhara and Oromia regional Bureaus of Health ethics committees.

### 2.4. Data Analyses

All analyses were conducted in Stata/SE 16.1.

Descriptive analyses. Descriptive statistics were calculated to describe the characteristics of the overall sample, polyvictims, and non-polyvictimized adolescents on the mediator (resilience), the dependent variables of interest (physical health and mental health), and all control variables, stratified by sex. The prevalence of experiencing individual forms of victimization and polyvictimization, stratified by sex, is also presented.

Regression analyses. Separate ordinary least squares regression models were run to assess the impact of polyvictimization on mental health (continuous outcome using a linear regression model) and physical health (binary outcome using the linear probability model). The linear probability regression results are consistent with logistic regression model results, but linear probability models are used for ease of interpretation. The regressions were stratified by sex (girls versus boys). First, in model 1, the unadjusted association between mental or physical health and polyvictimization at round two was assessed. Additional covariates were progressively added in a series of regression models. In model 2, mental or physical health were regressed on polyvictimization at round two, including demographic covariates (adolescent age, adolescent disability status, orphan status, household size, living in a female-headed household, household head illiteracy, household poverty, and urban versus rural location). In model 3, parental drug/alcohol use was added as an additional covariate. In model 4, gender norms and attitudes were added. These covariates were included in additional models because based on the literature, we expected these variables to independently predict the outcome. Finally, in model 5, the experience of polyvictimization at baseline was added.

The equation for the full model 5 exploring the association between polyvictimization and mental or physical health outcomes is
(1)$$y_{i}=\alpha +\beta_{1}PV_{i}+I_{i}+H_{i}+C_{ic}+{\beta_{0}PV}_{i}+\varepsilon_{ic}$$
where $y_{i}$ represents the outcome of interest, mental or physical health, for adolescent *i*, and $\beta_{1}PV_{i}$ is the term for polyvictimization at round two; the coefficient we are most interested in is $\beta_{1}$. $I_{i}$ represents a vector of individual-level covariates that were hypothesized to be associated with the dependent variable; $H_{i}$ represents a vector of household-level covariates; and $C_{ic}$ represents a vector of community-level covariates including community gender norms and location. The coefficient $\beta_{0}$ represents a control for whether or not the adolescent was polyvictimized at baseline. The standard errors, $\varepsilon_{ic},$ are clustered at the community level using the vce command in Stata to account for the clustered nature of the data [95].

Next, the impact of seven individual victimization types on mental and physical health were assessed, controlling for all other types of victimization. Adolescents who report having experienced any one of the items for a particular type of victimization (e.g., if they endorse one of the emotional violence questions) were considered to have experienced that type of victimization. Beta coefficients, *p*-values and analytic sample sizes are reported. Parameter estimates that are statistically different than zero are starred with three stars for *p* < 0.01, two stars for *p* < 0.05, and one star for *p* < 0.10.

Mediation. We used path analysis using the sem command in Stata with the maximum likelihood estimation model option to explore whether resilience mediates the relationship between polyvictimization and mental health and between polyvictimization and physical health. The sem command facilitates the calculation of all direct, indirect, and total effects in the path analysis, conceptually following the same series of steps described in Baron and Kenny [96], but allowing these effects to be calculated with a single command instead of carrying out separate individual regressions [96,97]. The direct effect represents the effect of polyvictimization on mental health or physical health, separately, after controlling for the mediator. Our hypothesis is that the indirect effect (ab) of polyvictimization on mental health or physical health through resilience would be statistically significant, meaning that path c′ (the direct effect of polyvictimization on mental health or physical health) would be a reduced estimate of c (the total effect of polyvictimization on mental health or physical health), indicating partial mediation. If the coefficient for path c′ was zero, this would indicate full mediation, which we did not hypothesize (see Figure 3).

The equation below assesses if being polyvictimized is associated with the mediator, resilience (path *a* in Figure 3 below). The coefficient $\beta_{3}$ represents the direct effect of polyvictimization on resilience.
(2)$$y_{i}=\alpha +\beta_{3}PV_{i}+I_{i}+H_{i}+C_{ic}+{\beta_{0}PV}_{i}+\varepsilon_{ic}$$

The next equation assesses if the mediator, resilience, is significantly associated with the outcomes (path b in Figure 3 below) and if the path from polyvictimization to mental health (or physical health) is reduced after controlling for resilience (i.e., if the direct effect, c′ in Figure 3 below, is reduced after taking ab, the indirect effect, into account). The equation for this regression is
(3)$$y_{i}=\alpha +\beta_{4}PV_{i}+\beta_{5}M_{i}+I_{i}+H_{i}+C_{ic}+{\beta_{0}PV}_{i}+\varepsilon_{ic}$$

The coefficient $\beta_{4}$ represents path c′ in Figure 3, or the direct effect of polyvictimization on poorer mental or physical health, through resilience. The coefficient $\beta_{5}$ represents path b in Figure 3, or the effect of the hypothesized, resilience, on poorer mental or physical health. The indirect effect, or the amount of mediation, is the product of paths a and b. The proportion of the total effect that is mediated is reported in the sem output and can also be calculated by taking the product of a and b and dividing it by the sum of ab plus the direct effect, c′.

For variables whose values were missing for adolescents who could not have been exposed to a particular type of violence due to personal characteristics, missing data were imputed as zero. For example, adolescents who were not enrolled in school could not have been exposed to forms of school-based violence. For the six items on peer victimization, 43 participants had missing data because these data were not collected at the time of data collection in Oromia region. We used inverse probability weighting (IPW) to account for these missing data since missing data were due to non-random lack of information. We used a probit model to estimate the weights, including adolescent age, location, and never-married status. The IPW weights were calculated as one divided by the estimated probability of having non-missing peer victimization data. Robust standard errors were used to adjust for estimation of the weights. A total of 241 eligible adolescents were missing data on the total resilience score. Some of these adolescents were skipped because they reported a different age during their survey than the adult female respondent, resulting in them being skipped for the resilience questions, which were asked of the younger cohort (ages 10–12) adolescents only. Other adolescents responded refused or do not know to at least one of the scale items, which led to incomplete responses on the 12-item summative scale. To address this missingness, adolescents who were missing four or more items were dropped from the sample used for the mediation analyses only in order to maximize the available sample size for other analyses; this resulted in 43 adolescents being dropped. For adolescents who were only missing one, two, or three items on the scale, mean values for those individual items were imputed (for a total of 198 respondents), and then the resilience scale was recalculated.

We ran additional path analyses to assess whether resilience mediates the associations between sexual violence and mental or physical health, and whether resilience mediates the associations between witnessing violence and mental or physical health among adolescent girls and boys, using the same procedures described above. The results of these supplemental analyses are presented in Appendix B.

## 3. Results

### 3.1. Descriptive Statistics

Descriptive statistics of victimization experiences by sex are presented in Table 3. Half of females (50%) and more than half of males (53.49%) in the sample experienced polyvictimization in the last year. Among females, the most common type of individual victimization reported was FGM/C, followed by household emotional violence, school-based violence, peer violence, household physical violence, witnessing violence, and then sexual violence. Among males, the most common type of individual victimization reported was school-based violence, followed by household emotional violence, peer violence, household physical violence, witnessing, and sexual violence. Table 4 displays descriptive statistics for the dependent variables, mental health and physical health, the mediator, resilience, and other control variables among females and males in the sample, stratified by victimization status. Overall, the descriptive analyses suggest that mean mental health and physical health are relatively high across the overall sample and subgroups. Mean resilience is also relatively high in the overall sample and subgroups.

### 3.2. Regression Analyses

All regression analysis results on mental health are provided first, followed by the regression analysis results on physical health. We also examined the association of cumulative polyvictimization (i.e., the experience of polyvictimization at either baseline and/or round two) on mental health and physical health, and the results were similar to the results when examining the association between polyvictimization at round two only and adolescent outcomes (results available upon request). Table 5 presents the regression results of the association between polyvictimization and mental health. We see significant associations between polyvictimization and mental health for both females and males. For females (Panel A), being polyvictimized was associated with a 0.284 unit increase in mental distress (*p* < 0.01), after adjusting for all covariates in the fifth and final regression model. For males (Panel B), after adjusting for all covariates in the fifth and final regression model, being polyvictimized was associated with a 0.357 unit increase in mental distress (*p* < 0.01).

Table 6 presents the association between individual forms of victimization and mental health among females and males (Panels A and B, respectively), controlling for other forms of victimization. Among females, the experience of sexual violence, witnessing, and peer violence were significant at the *p* < 0.10 level or below after adjusting for the full set of covariates. Among males, witnessing violence against one’s mother/female guardian, experiencing household physical violence, experiencing household emotional violence, and experiencing peer violence were each significantly associated at the *p* < 0.10 level with increased mental distress after adjusting for the full set of covariates.

Next, we regressed polyvictimization on physical health. There were no significant associations between polyvictimization and physical health for either females or males in any of the models (Table 7).

When we regressed the experience of individual forms of victimization on adolescent physical health for females and for males, controlling for the experiences of any other type of victimization in the unadjusted regressions and for the full set of covariates in the adjusted regressions, only the experience of peer violence was significantly associated with reporting poor or very poor health among both sexes (Table 8). Peer violence was associated with a 0.071 unit increase in the likelihood of reporting poor or very poor physical health among girls (*p* < 0.01), and with a 0.039 unit increase among boys (*p* < 0.01), after controlling for all covariates.

### 3.3. Mediation

We first examined whether resilience mediated the association between polyvictimization and mental distress among adolescent girls and boys, separately, controlling for the full set of covariates used in the regression analyses above. The results of these path analyses are shown in Figure 4 and Figure 5, with the standardized coefficients and robust standard errors reported for each path.

Among girls, polyvictimization had a direct, negative association with resilience as hypothesized (path a, β = −0.048, *p* < 0.10) and a direct, positive association with poorer mental health (path c′; β = 0.084, *p* < 0.01) (Figure 4). As hypothesized, resilience had a direct, negative association with poorer mental health (path b, β = −0.204, *p* < 0.01), meaning that resilience plays a direct role in mitigating or buffering poorer mental health, by reducing poorer mental health outcomes (higher mental health scores indicate poorer mental health). The indirect effect, ab, of polyvictimization on increased mental distress, our coefficient of interest for the path analysis, was also statistically significant, but only at a 0.10 significance level (β = 0.010, *p* < 0.10), indicating that the relationship between polyvictimization and poorer mental health is mediated by resilience among adolescent girls. This means that polyvictimization decreases mental health through reducing resilience.

Among boys, contrary to our hypothesis, the direct effect of polyvictimization on resilience was not significant (path a, β = −0.004, *p* = 0.894), but polyvictimization did have a direct, positive association with poorer mental health (path c′; β =0.113, *p* < 0.01) (Figure 5).

As hypothesized, resilience had a direct, negative association with poorer mental health (path b, β = −0.189, *p* < 0.01). The indirect effect, ab, of polyvictimization on increased mental distress, was not significant (β =0.001, *p* = 0.894, indicating that the relationship between polyvictimization and poorer mental health is not mediated by resilience among males. We ran additional path analyses to explore whether resilience mediated the associations between sexual victimization and mental health, or between witnessing violence against the mother/female guardian and mental health. The results of these mediation analyses are presented in Appendix B.

We then examined whether resilience mediated the association between polyvictimization and physical health among adolescent girls and boys, separately, controlling for the full set of covariates used in the regression analyses above. Contrary to our hypothesis, we did not find evidence for mediation, because the indirect effect ab of polyvictimization on poorer physical health was not significant among either girls or boys. The results of these mediation analyses are presented in Appendix B. We also ran additional path analyses to explore whether resilience mediated the associations between sexual victimization and physical health, or between witnessing violence against the mother/female guardian and physical health. The results of these mediation analyses are presented in Appendix B.

## 4. Discussion

This is the first study we are aware of to explore the relationships between polyvictimization and adolescent mental and physical health outcomes including the mediating role of resilience among adolescents in an LMIC setting. The results indicate that polyvictimization, a universal issue among adolescents and defined as the experience of two or more forms of victimization, was very common among this sample of Ethiopian adolescents, and is experienced by approximately half of adolescent girls and over half of adolescent boys. Polyvictimization had a significant, adverse effect on adolescent self-reported mental health, similar to previous studies examining this association [10,52], although unlike the previous literature, we found that the association was more pronounced among boys than among girls [52]. However, contrary to our hypothesis, and to some of the previous literature examining this association [53,98], polyvictimization was not significantly associated with worse self-reported physical health. Future longitudinal studies assessing self-reported physical health after experiencing polyvictimization may be better able to detect an association between polyvictimization and adverse physical health outcomes. Interestingly, among both girls and boys, the only individual type of victimization experience that independently had an association with worse physical health was peer victimization, underscoring the consequential role that peer relations have on adolescent well-being as the previous literature has demonstrated [99,100].

By contrast, in addition to peer violence, sexual violence, witnessing violence against the adolescent’s mother/female guardian, and household physical violence were all independently associated with worse mental health among girls, after controlling for other forms of victimization and adjusting for all covariates. Contrary to our hypothesis, FGM/C was not associated with more adverse physical or mental health among girls; it is possible that in the Ethiopian setting, where FGM/C is normative, FGM/C could potentially be protective for some if being uncircumcised is associated with stigma due to social norms and expectations to undergo FGM/C [101].

Among boys, peer violence, witnessing violence, household physical violence, and household emotional violence were all independently associated with worse mental health, after controlling for other forms of victimization and adjusting for all covariates. The lack of a significant association between experiencing sexual victimization and adverse mental health among boys may be because many fewer boys reported experiencing sexual violence compared to girls, indicating a gender difference in the risk of experiencing sexual violence [34]. However, there may also be under-reporting among boys due to greater stigma. The finding that experiencing forms of school-based violence, including being physically or emotionally abused or punished by teachers, was not associated with adverse mental health was also unexpected and contrary to our hypothesis and the previous literature [28]. The finding that witnessing violence against the adolescent’s mother/female guardian had the greatest adverse association with mental health compared to polyvictimization or other direct, individual forms of victimization among both girls and boys was unexpected. These results underscore the urgent need for further research to understand why witnessing violence against one’s mother or female guardian/caregiver is so detrimental to adolescent mental health and the characteristics associated with these experiences (such as frequency, types of acts witnessed, etc.).

Resilience significantly mediated the association between polyvictimization and mental health among adolescent girls, but not among adolescent boys, indicating partial support for our hypothesis. There may be several potential reasons why resilience was a significant mediator for girls, but not for boys. It is possible that resilience may operate differently among girls and boys, and according to developmental age. There is evidence for sex differences in resilience to stress, with girls who experience stress exposures such as polyvictimization and other adverse experiences during or post puberty being more likely to experience stronger proximal adverse effects of stress (e.g., anxiety, depression, PTSD) than girls who experience stress when they are pre-pubescent [102]. Longitudinal studies show that women who experience an ACE in the pre-pubertal period are more resilient to experiencing depression compared with those who experience adversities during or post-puberty, and women who did experience a childhood adversity in the prepubertal window were more resilient to depression as adults, even if they later experienced additional adversities in the post-pubertal period [102,103]. Since the study sample of younger cohort adolescents included a mix of pre and post-pubertal girls and boys, the timing of their exposures to polyvictimization may have also impacted their resilience. Overall, these study findings point to the need for future research, including qualitative research, for a more in-depth exploration of polyvictimization, its associations with adolescent mental and physical health, and resilience in order to understand individual-level experiences, processes, and perspectives [104].

This study has several limitations. GAGE does not ask about lifetime victimization for any victimization category except for experiencing school-based violence and sexual violence; all other victimization questions ask about the previous 12 months only, and so we will not be able to assess earlier experiences of victimization over the adolescent’s entire lifetime. The GAGE survey did not include items covering all forms of victimization. For example, neglect was not assessed, nor was the experience of IPV or dating violence. The lack of data on these specific forms of victimization may have (1) contributed to under-reporting of polyvictimization and (2) limited exploration of how polyvictimization is associated with the adolescent well-being dependent variables. Second, is possible there were unobserved variables not measured or included in the analysis that were correlated with the predictor and influenced the dependent variables, leading to confounding. For example, we do not have any data on the experience of dating violence to include in our measure of polyvictimization, although this experience might be minimal in this younger sample of adolescents. Using theory and the literature to select covariates to include in the regression models may have helped to reduce, but not eliminate, the threat of omitted variable bias [105]. Third, the GAGE survey relies on self-reported data, which may be influenced by recall bias and/or telescoping bias [106], social desirability bias, and under-reporting of sensitive or socially unacceptable experiences or behaviors. Finally, overall, causal inference in this observational, cross-sectional design is not possible, and all reported interpretations should be treated as associational in nature.

Including witnessing domestic violence in the polyvictimization is worth further discussion. It is clear that witnessing domestic or intimate partner violence has adverse consequences for adolescents such as increasing the risk of externalizing behaviors including perpetration of violence (particularly among boys); internalizing issues such as depression and anxiety; reduced attachment to parents among adolescents of both sexes [107,108]; and an increased risk of being in an abusive relationship among girls [109,110]. However, there is a debate in the field over whether or not to include witnessing domestic violence, regardless of its negative impacts, in definitions of polyvictimization. This debate stems from the potential risk of blaming mothers and mothers losing child custody in the context of failure to protect-related legislation and policies, even when fathers are more likely to be the perpetrators [19,20]. While acknowledging that such victim blaming and stigmatization is harmful and can reduce reporting and help-seeking among women [111], this study includes witnessing domestic violence as a type of indirect form of victimization in its examination of polyvictimization. However, as suggested by other scholars, categorizing witnessing domestic violence as a form of victimization may be confounding cause with effect [21]. While this paper does not resolve this debate, it contributes to disentangling how exposure to domestic violence impacts adolescent mental health, independent of other forms of victimization that often co-occur such as household physical and emotional violence, and other ACEs included in the study, such as poverty.

## 5. Conclusions

This study begins to address key gaps in the literature on understanding the impacts of polyvictimization among adolescents in LMICs, the extent to which polyvictimization experiences are gendered, and the mediating role of resilience. To date, most polyvictimization research has taken place in HICs, has not included all forms of violence that adolescents in LMICs are likely to experience such as FGM/C and school violence, and has focused on mental health outcomes with very limited evidence on physical health outcomes [10]. In settings such as Ethiopia where exposure to victimization is high, understanding the pathways by which polyvictimization influences adolescent outcomes and capabilities can help inform more effective policies and targeted programming to intervene during adolescence, a critical developmental period.

This study provides evidence that girls and boys experience violence differentially, both in terms of the types of violence that they are more likely to experience, and the associations these experiences have with mental health, pointing toward the need for more gender-specific analyses. The evidence that polyvictimization adversely impacts adolescent mental health, and that, among girls, resilience mediates the association between polyvictimization and mental health, suggests the importance of intervening early to prevent victimization and to strengthen resilience in order to mitigate adverse consequences of polyvictimization, for example, through evidence-based social norms community mobilization programming such as SASA! [112,113] and life skills programming and social protection programs targeting both adolescents and their parents [114,115]. In addition, there is a need for further research, particularly in LMICs, on effective psychotherapeutic and other interventions to improve outcomes for victimized adolescents, and, in particular where there is co-occurring domestic violence, psychotherapeutic protocols that take a dyadic approach including the adolescent and non-abusive parent [116,117,118]. Research findings should be used to address intersecting forms of violence against adolescents and violence against women through increased prevention and intervention resources, adequate legal frameworks, changing social and gender norms about the acceptability of violence, and implementing integrated, multisectoral approaches to ending such violence [119].

Given that adolescent well-being is predictive of adulthood health and mental health outcomes and economic participation [120,121,122], increased attention to polyvictimization and effective prevention and treatment efforts can positively shift adolescents’ life trajectories and contribute to achieving broader developmental goals in LMIC contexts such as Ethiopia.

## Figures and Tables

**Figure 1 ijerph-20-06755-f001:**
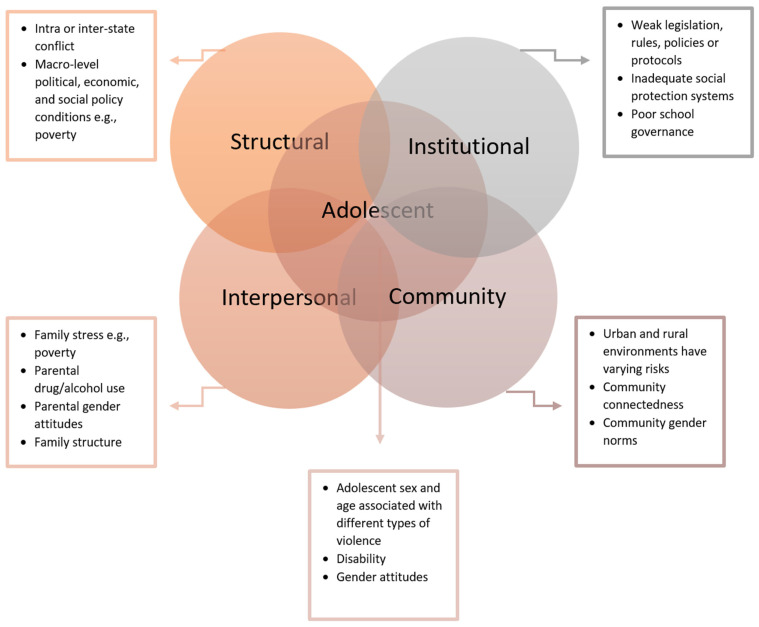
The Integrated Framework for Addressing Violence Affecting Children. Adapted from the Child-Centered Integrated Framework for Violence Prevention [43,64,66].

**Figure 2 ijerph-20-06755-f002:**
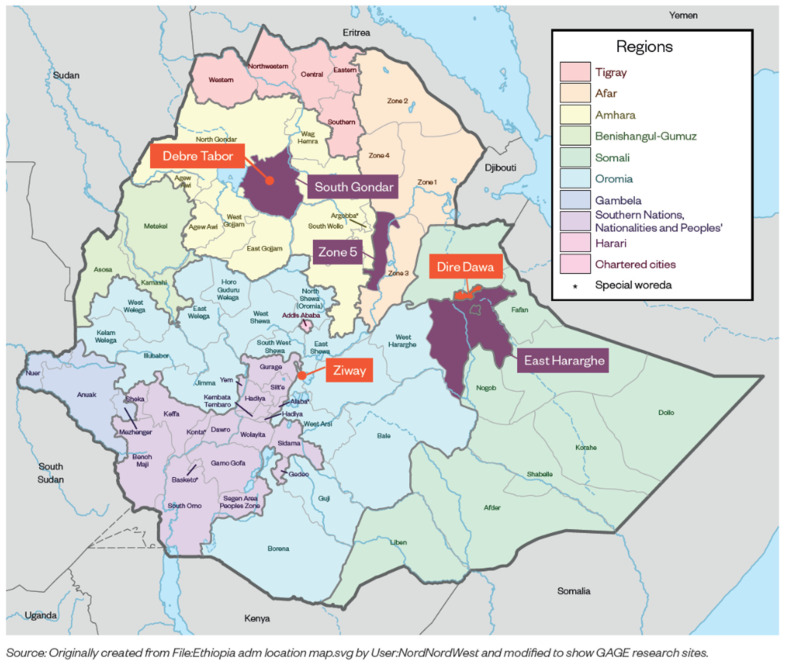
Map of GAGE research sites in Ethiopia; rural research sites are labeled in purple and urban research sites are labeled in orange [69].

**Figure 3 ijerph-20-06755-f003:**
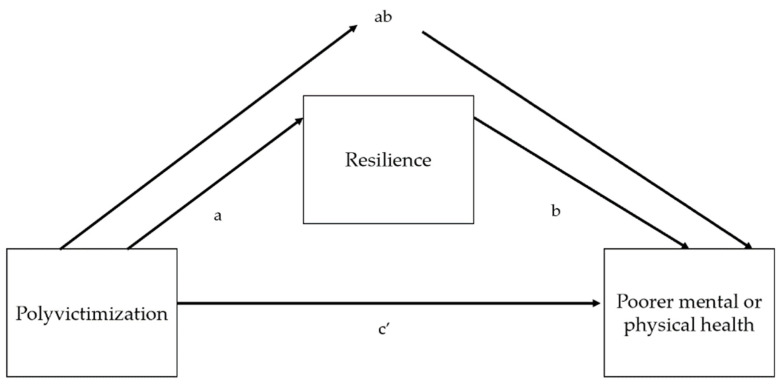
Mediation model of resilience, polyvictimization, and mental or physical health. Note that for visual clarity, control variables are omitted. Paths ab = indirect effect of polyvictimization on poorer mental or physical health through resilience. Path c′ = direct effect of polyvictimization on poorer physical or mental health. Term a × b + c′ = total effects.

**Figure 4 ijerph-20-06755-f004:**
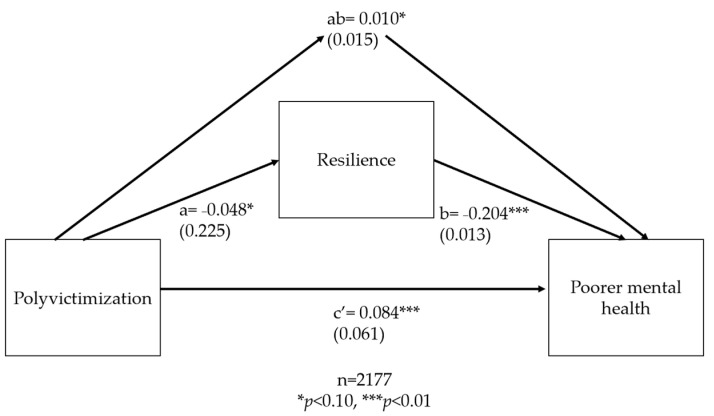
Path analysis of resilience, polyvictimization, and mental health among adolescent females. Notes: Standardized coefficients are reported with robust standard errors in parentheses.

**Figure 5 ijerph-20-06755-f005:**
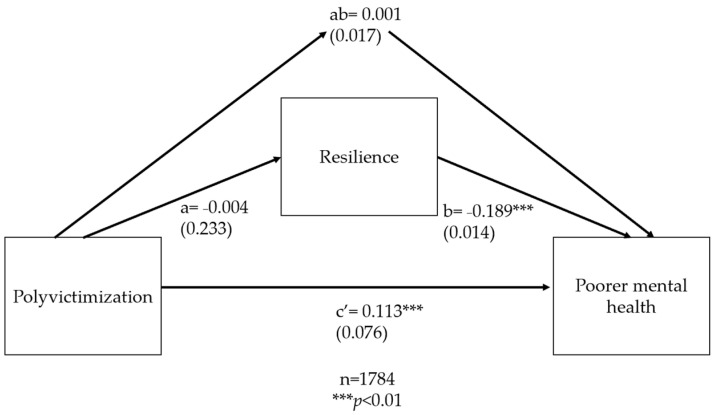
Path analysis of resilience, polyvictimization, and mental health among adolescent males. Notes: Standardized coefficients are reported with robust standard errors in parentheses.

**Table 1 ijerph-20-06755-t001:** Types of violence ^1^.

Type of Violence	Survey Response Options	Variable Construction	Number of Survey Items Used to Construct Variable
Household physical violence	Never happened, happened once, and happened more than once	Dichotomized to 1 if happened once or more than once reported or 0 if never happened	One
Household emotional violence	Never happened, happened once, and happened more than once	Dichotomized to 1 if happened once or more than once reported on either item or 0 if never happened on all items	Two
Sexual violence	Yes, no, refused, or don’t know Happy face, sad face, or no image chosen	Dichotomized to 1 if yes reported on either item or 0 if no, refused, or don’t know on all items Dichotomized to 1 if sad face chosen or 0 if happy face or no image chosen	Two ^2^ One ^3^
Witnessing violence against mother/female guardian	Never happened, happened once, and happened more than once	Dichotomized to 1 if happened once, more than once, or refused reported on either item or 0 if never happened on all items	Two
Peer violence	Never, once, or more than once	Dichotomized to 1 if happened once or more than once or 0 if never	Six
School violence	Yes, no, refused, or don’t know	Dichotomized to 1 if yes or 0 if no, refused, or don’t know	Three
FGM/C	Yes, no, refused, or don’t know	Dichotomized to 1 if yes or refused, or 0 if no or don’t know	Two

^1^ Items come from round two of the GAGE Ethiopia adolescent and adult female modules available at https://www.gage.odi.org/publication/ethiopia-round-2-survey-2019-2020/ (accessed on 3 August 2020). ^2^ These two violence items were asked of girls only. ^3^ Both girls and boys were given the option to non-verbally respond about their experience of sexual violence by marking a card with images of sad and happy faces.

**Table 2 ijerph-20-06755-t002:** Description of covariates used in analyses ^1^.

Level	Covariate	Survey Response Options	Variable Construction
Individual level	Adolescent age	Numeric	Continuous
	Biological sex	Female or male	Female = 1, male = 0
	Disability status	Either yes, no, refused, or don’t know options or no difficulty, some difficulty, a lot of difficulty, or cannot do at all.	Questions covered six domains of functioning including seeing, hearing, walking, self-care, cognition, and communication (understanding and being understood). Adolescents who responded with no, refused, don’t know, or some difficulty in all six domains were 0, no disability. Adolescents who responded with either a lot of difficulty or cannot do at all to at least one of the six domains were 1, have a disability.
	Orphan status (one or both parents deceased)	Yes, no, refused, or don’t know	0 (no, refused, or don’t know) indicating that neither the biological father or biological mother were reported to have died and 1 (yes) if either the biological father or mother were reported to have died.
	Gender attitudes	Agree, partially agree, disagree, refused, or don’t know	32 gender attitude questions covering domains of education, time use, financial inclusion and economic empowerment, relationships and marriage, and sexual and reproductive health. 1 = agree or partially agree, 0 = refused, or don’t know, or disagree, where agreement represents a gendered response and is reverse coded for items for which agreements suggests a nongendered response). Gender attitude scores were summed and standardized; higher scores reflect more inequitable gender attitudes and norms.
Household level	Household size	Numeric	Continuous variable for household size was calculated from the household roster of the adult female module by adding the number of adults and the number of children who were reported to typically live in the household.
	Living in a female-headed household	Male, female, refused, don’t know	Female = 1, male = 0, refused, or don’t know.
	Household head literacy	Yes, no, refused, or don’t know	Cannot read or write = 1, can read or write = 0, refused, or don’t know
	Household poverty	Whether the household owns a given item (yes, no, refused, or don’t know)	Asset deciles created using principal component analysis to capture material assets using an asset list from the adult female module [79,82]. Yes = 1, no = 0, refused, or don’t know.
	Parental alcohol or drug use	Yes, no, refused, not applicable, or don’t know	Dichotomized to yes = 1, no = 0, refused, or don’t know.
Community level	Gender norms	Agree, partially agree, disagree	Fourteen norms-related questions (Domains and scoring same as in gender attitudes).
	Type of location	Rural or urban	0 = rural, 1 = urban

^1^ Items come from round two of the GAGE Ethiopia adolescent and adult female modules.

**Table 3 ijerph-20-06755-t003:** Descriptive statistics of victimization experiences by sex in the analytic sample from Ethiopia ^1^.

Sex	Polyvictimized N (%)	Non-Polyvictimized N (%)	Household Physical Violence N (%)	Household Emotional Violence N (%)	Sexual Violence N (%)	Witnessing Violence N (%)	Peer Violence N (%)	School-Based Violence N (%)	FGC/M N (%)
Male	959 (53.49)	854 (47.10)	591 (32.60)	842 (46.62)	70 (3.86)	177 (9.76)	690 (38.48)	911 (50.44)	n/a
Female	1094 (50.00)	1094 (50.00)	493 (22.27)	775 (35.08)	158 (7.15)	199 (8.99)	491 (22.44)	715 (32.37)	1031 (46.57)
Overall	2053 (51.57)	1948 (48.93)	1084 (26.92)	1617 (40.27)	228 (5.67)	376 (9.34)	1181 (29.67)	1626 (40.50)	n/a

^1^ This table summarizes the main independent variables (polyvictimization and individual types of victimization) reported by adolescents in the GAGE survey.

**Table 4 ijerph-20-06755-t004:** Descriptive statistics of resilience, mental health, physical health, and control variables among adolescents in the analytic sample from Ethiopia ^1^.

**Panel A: Females**	**Female,** **Overall**			**Female,** **Non-Polyvictimized**			**Female,** **Polyvictimized**		
	**N**	**Mean**	**sd**	**N**	**Mean**	**sd**	**N**	**Mean**	**sd**
Resilience Score (higher scores = more resilient, max = 48)	2213	31.82	4.28	1094	32.21	4.11	1094	31.45	4.40
Poor health	2214	0.11	0.32	1094	0.11	0.31	1094	0.12	0.32
Score on GHQ-12 (0–12, higher = more distress)	2214	0.90	1.50	1094	0.71	1.30	1094	1.10	1.67
Age at R2 survey	2214	12.89	0.90	1094	12.93	0.89	1094	12.86	0.90
Has a disability	2213	0.05	0.21	1093	0.04	0.20	1094	0.05	0.22
1 or both parents are deceased	2214	0.09	0.29	1094	0.08	0.28	1094	0.10	0.29
Household head is female	2214	0.18	0.39	1094	0.17	0.37	1094	0.20	0.40
Household head illiterate	2214	0.63	0.48	1094	0.60	0.49	1094	0.65	0.48
Standardized gender attitudes index	2214	0.37	0.66	1094	0.30	0.66	1094	0.44	0.65
Standardized gender norms index	2214	0.10	1.00	1094	0.10	1.00	1094	0.11	1.01
Household size	2207	6.31	2.30	1089	6.24	2.34	1093	6.38	2.26
Either parent/guardian drinks alcohol or uses chat	2214	0.54	0.50	1094	0.48	0.50	1094	0.59	0.49
Standardized index of household assets (decile)	2203	5.19	2.99	1087	5.24	3.04	1091	5.14	2.95
Lives in urban area	2214	0.13	0.33	1094	0.11	0.32	1094	0.14	0.35
**Panel B: Males**	**Male,** **Overall**			**Male,** **Non-Polyvictimized**			**Male,** **Polyvictimized**		
	**N**	**Mean**	**sd**	**N**	**Mean**	**sd**	**N**	**Mean**	**sd**
Resilience Score (higher = more resilient, max = 48)	1809	32.33	3.79	343	31.53	4.64	959	32.24	3.56
Poor health	1807	0.08	0.27	343	0.08	0.27	959	0.08	0.27
Score on GHQ-12 (0–12, higher = more distress)	1810	1.02	1.55	343	0.80	1.30	959	1.20	1.71
Age at R2 survey	1813	12.98	0.90	343	13.05	0.94	959	12.95	0.91
Has a disability	1810	0.07	0.25	343	0.08	0.27	959	0.07	0.26
1 or both parents are deceased	1812	0.09	0.29	343	0.09	0.28	959	0.10	0.30
Household head is female	1813	0.16	0.36	343	0.19	0.39	959	0.16	0.37
Household head illiterate	1813	0.59	0.49	343	0.64	0.48	959	0.59	0.49
Standardized gender attitudes index	1813	0.49	0.62	343	0.59	0.64	959	0.49	0.61
Standardized gender norms index	1813	0.04	0.98	343	0.08	0.99	959	0.06	1.01
Household size	1804	6.11	2.21	341	5.74	2.44	954	6.20	2.19
Either parent/guardian drinks alcohol or uses chat	1813	0.50	0.50	343	0.38	0.49	959	0.56	0.50
Standardized index of household assets (decile)	1804	5.41	2.84	341	5.40	2.78	954	5.29	2.87
Lives in urban area	1813	0.17	0.38	343	0.11	0.31	959	0.20	0.40

^1^ This table summarizes the independent variables, mediator, and dependent variables. There are small differences in sample sizes across outcomes. The maximum sample size for each subsample is indicated in this table in the first column of each subsample (N). The exact sample size for each analysis is presented in the regression tables.

**Table 5 ijerph-20-06755-t005:** Association of polyvictimization with mental health ^1^.

	Score on GHQ-12 (0–12, Higher = More Mental Distress)									
	Panel A: Females					Panel B: Males				
	Model 1	Model 2	Model 3	Model 4	Model 5	Model 1	Model 2	Model 3	Model 4	Model 5
Polyvictimized	0.386 *** (0.067)	0.338 *** (0.067)	0.301 *** (0.066)	0.298 *** (0.066)	0.284 *** (0.064)	0.377 *** (0.079)	0.359 *** (0.077)	0.367 *** (0.079)	0.361 *** (0.078)	0.357 *** (0.079)
Sample size	2188	2177	2177	2177	2177	1793	1784	1784	1784	1784

^1^ Regression coefficients are reported, and standard errors are reported in parentheses. Parameter estimates are statistically significant at *** *p* < 0.01. Model 1 is the unadjusted model. Model 2 includes adolescent age, disability, orphan status, living in a female-headed household, househod head illiteracy, household size, household assets, and living in an urban area as control variables. Model 3 includes control variables in Model 2 in addition to parental drug or alochol use. Model 4 includes all control variables in Models 2–3, in addition to gender attitudes and gender norms. Model 5 includes all preceding control variables, in addition to polyvictimization at baseline.

**Table 6 ijerph-20-06755-t006:** Association of victimization with self-reported mental health ^1^.

	Score on GHQ-12 (0–12, Higher = More Mental Distress)			
	Panel A: Females		Panel B: Males	
	Unadjusted Coefficient	Adjusted Coefficient	Unadjusted Coefficient	Adjusted Coefficient
Sexually victimized	0.450 *** (0.158)	0.439 *** (0.146)	0.390 (0.278)	0.351 (0.254)
Witnessed violence against mother/female guardian	0.530 *** (0.152)	0.507 *** (0.146)	0.803 *** (0.189)	0.695 *** (0.178)
Experienced school violence	0.076 (0.072)	0.189** (0.084)	−0.113 (0.087)	1793
Experienced household physical violence	0.252 *** (0.091)	0.123 (0.081)	0.392 *** (0.105)	0.347 *** (0.097)
Experienced household emotional violence	0.149 * (0.086)	0.070 (0.063)	0.225 ** (0.095)	0.204 ** (0.094)
Experienced FGM/C	0.147 ** (0.073)	0.345 *** (0.087)		
Experienced peer violence	0.378 *** (0.089)	0.439 *** (0.146)	0.216 ** (0.102)	0.233 ** (0.099)

^1^ Regression coefficients are reported, and standard errors are reported in parentheses. Control variables for all other potential types of victimization were included in both analyses. Control variables for adolescent age, disability, having lost one or both parents, living in a female-headed household, household head illiteracy, household size, household asset decile, living in an urban versus rural area, parental alcohol or chat use, gender attitudes, gender norms, all other types of victimizations experienced, and baseline victimization were included in the adjusted coefficient results. Sample size for unadjusted regression is n = 2188 for females and 1793 for males. Sample size for adjusted regression is n = 2177 for females and 1784 for males. Parameter estimates are statistically significant at * *p* < 0.10, ** *p* < 0.05, *** *p* < 0.01.

**Table 7 ijerph-20-06755-t007:** Association of polyvictimization with physical health ^1^.

	=1 if Physical Health is Poor or Very Poor									
	Panel A: Females					Panel B: Males				
	Model 1	Model 2	Model 3	Model 4	Model 5	Model 1	Model 2	Model 3	Model 4	Model 5
Polyvictimized	0.009 (0.015)	0.010 (0.015)	0.013 (0.014)	0.016 (0.014)	0.014 (0.014)	−0.023 (0.019)	−0.0157 (0.018)	−0.013 (0.018)	−0.013 (0.018)	−0.001 (0.020)
Sample size	2215	2204	2204	2204	2204	2215	1789	1789	1789	1789

^1^ Regression coefficients are reported, and standard errors are reported in parentheses. Model 1 is the unadjusted model. Model 2 includes adolescent age, disability, orphan status, living in a female-headed household, househod head illiteracy, household size, household assets, and living in an urban area as control variables. Model 3 includes control variables in Model 2 in addition to parental drug or alochol use. Model 4 includes all control variables in Models 2–3, in addition to gender attitudes and gender norms. Model 5 includes all preceding control variables, in addition to polyvictimization at baseline.

**Table 8 ijerph-20-06755-t008:** Association of victimization with physical health ^1^.

	=1 if Physical Health is Poor or Very Poor			
	Panel A: Females		Panel B: Males	
	Unadjusted Coefficient	Adjusted Coefficient	Unadjusted Coefficient	Adjusted Coefficient
Sexually victimized	−0.004 (0.025)	−0.005 (0.023)	0.035 (0.038)	0.037 (0.036)
Witnessed domestic violence	0.022 (0.028)	0.029 (0.027)	−0.013 (0.021)	−0.010 (0.021)
Experienced school violence	−0.013 (0.015)	0.021 (0.015)	−0.008 (0.010)	−0.010 (0.012)
Experienced household physical violence	0.028 (0.018)	0.035 (0.018)	−0.007 (0.013)	0.007 (0.012)
Experienced household emotional violence	−0.011 (0.018)	−0.012 (0.016)	0.007 (0.013)	0.011 (0.013)
Experienced FGM/C	−0.024 (0.015)	−0.016 (0.014)		
Experienced peer violence	0.082 *** (0.021)	0.071 *** (0.020)	0.045 *** (0.013)	0.039 *** (0.013)

^1^ Regression coefficients are reported, and standard errors are reported in parentheses. A control variable for all other potential types of victimization was included in the unadjusted coefficient results. Control variables for adolescent age, disability, having lost one or both parents, living in a female-headed household, household head illiteracy, household size, household asset decile, living in an urban versus rural area, parental alcohol or chat use, gender attitudes, gender norms, baseline victimization were included in the adjusted coefficient results. Sample size is n = 2188 for females and n = 1791 for males in the unadjusted regressions. Sample size is n = 2177 for females and n = 1782 for males in the adjusted regressions. Parameter estimates are statistically significant at *** *p* < 0.01.

## Data Availability

The data presented in this study are openly available in the UK Data Service at https://doi.org/10.5255/UKDA-SN-8597-1 (accessed on 10 March 2021). Due to sensitivity, data on victimization presented in this study are available on request from S.B.

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
