# Peer review of "Polyvictimization and Adolescent Health and Well-Being in Ethiopia: The Mediating Role of Resilience"

_ijerph, 2023, doi:10.3390/ijerph20186755_

Round 1

Reviewer 1 Report

 I am grateful to be able to analyze at first hand such an elaborate research and with such a relevant purpose.

Below I present elements that, from my point of view, can help the authors to achieve their purpose.

Introduction

It is essential to insert references about the impacts of violence on a person's (physical and mental) health. It is also fundamental to define health (physical and mental); I recommend using the WHO definitions as they are globally recognized and agreed upon based on scientific evidence.

Research Design

There are correlations that seem statistically correct but conceptually problematic. Let me explain: the statement that the impact of polyvictimization does not have a significant index on physical health is incorrect, since the research works with self-reported physical health, and not with direct health indicators.

The article is dedicated to self-reported health, that is, it works with a perception anchored in notions established by dominant coercive discourse, and, therefore, self-perception may be under the effort of meeting the demands of what is posed by dominant groups as correct – the case of FGM/C “sold” as an action for the benefit of women, when it is not (women from these countries with training in the health area have denounced this situation). The self-perception of health constrained by dominant coercive discourses works as self deceit – research on the telomere effect, for example, suffered in the bodies of people who live systematically subjected to violence reveals the direct impact on people's health.

This limitation has to be described directly in the presentation of the research design.

Results

Due to the elements exposed above, the results will be more relevant if the central questions of the study are conceptually problematized. The same must be done in the conclusions.

By not addressing the issues of naturalization of violent practices in the context studied, the article oscillates between reinforcing this naturalization and generalizing the idea that violence is at the base of low-income countries/cultures.

Data are important, but their presentation and analysis need greater articulation with issues of violence, dominant coercive discourse processes and evidence with social impact to overcome these practices. This is a challenge posed to research in general, but especially those that focus on non-European cultural contexts. It is also a challenge for quantitative research that can contribute a lot to overcoming inequalities and violence but for which it has to deepen conceptual issues.

Author Response

Dear Reviewer,

Thank you for the valuable comments, feedback, and guidance, which we have taken into account in the revised version of the paper. We think that these revisions have led to considerable improvement in the paper, and hope that we have been able to address all of your comments satisfactorily.

Best, Lior Miller (on behalf of the co-authors).

Reviewer 2 Report

The topic addressed by the authors entitled: "Polyvictimization and Adolescent Health and Well-Being in 2 Ethiopia: The Mediating Role of Resilience" refers to issues of great importance for contemporary research.

As for its strong point, it stands out the fact that it analyzes a phenomenon in which it is necessary to deepen.

• The manuscript provides information relevant to the topic and this is also a strong point. Likewise, it should be noted that some recent references have been used together with other more classic ones, which helps to see the evolution of the terms.

• In addition, the authors correctly identified the topic as the objectives of the research and the proposed hypothesis.

• The methodological assumptions used in this project are unquestionable. The results of the study are clearly presented. There are also no doubts about the interpretation of the results.

• And the “discussion” adjusts to the appropriate scientific parameters, in which its results are contrasted with other previous ones.

• The authors indicate that it is based on a project, as well as the ethical criteria that they have followed to develop the research.

• And, finally, the applied bibliographical references are correct.

Regarding aspects to improve, it is recommended to make substantial changes:

• Title: Following the APA regulations, titles should not contain more than twelve words, it is recommended to limit the length of the title.

• The summary does not provide all the information, it is recommended to follow the structure: Introduction, empirical development, results and conclusion.

• Although the theoretical framework is well developed, a greater structure is recommended (they must provide various subpoints that give greater meaning and structure to the literature of the study) and expansion of it.

• The authors do not provide “limitations of the study or future lines of research”, which will give their study greater scientific rigor.

Congratulations to the authors on their study, and I encourage them to make these minimal changes that will give their study greater rigor.
